# Liquid Metal Flow Under Traveling Magnetic Field—Solidification Simulation and Pulsating Flow Analysis

**Evgeniy Shvydkiy** [1,*] **, Egbert Baake** [2] **and Diana Köppen** [2]

1   Department of Electrical Engineering and Electrotechnology Systems, Ural Federal University, 620078 Yekaterinburg, Russia
2   Institute of Electrotechnology, Leibniz Universität Hannover, 30167 Hannover, Germany; baake@etp.uni-hannover.de (E.B.); diana.koeppen@lenze.com (D.K.)
*   Correspondence: e.l.shvydky@urfu.ru; Tel.: +7-343-375-95-14

**Abstract:** Non steady applied magnetic field impact on a liquid metal has good prospects for industry. For a better understanding of heat and mass transfer processes under these circumstances, numerical simulations are needed. A combination of finite elements and volumes methods was used to calculate the flow and solidification of liquid metal under electromagnetic influence. Validation of numerical results was carried out by means of measuring with ultrasound Doppler velocimetry technique, as well as with neutron radiography snapshots of the position and shape of the solid/liquid interface. As a result of the first part of the work, a numerical model of electromagnetic stirring and solidification was developed and validated. This model could be an effective tool for analyzing the electromagnetic stirring during the solidification process. In the second part, the dependences of the velocity pulsation amplitude and the melt velocity maximum value on the magnetic field pulsation frequency are obtained. The ability of the pulsating force to develop higher values of the liquid metal velocity at a frequency close to the MHD resonance was found numerically. The obtained characteristics give a more detailed description of the electrically conductive liquid behaviour under action of pulsating traveling magnetic field.

**Keywords:** electromagnetic stirring; forced convection; traveling magnetic field; liquid metal; solidification; numerical analysis; pulsed magnetic field; gallium

## 1. Introduction

Alternating electromagnetic (EM) field impact on electrically conductive liquid generates Lorenz forces within its volume and leads to fluid movement. Thus, the magnetic field acts as a stirring device and it can, in principle, be engineered to provide any desired pattern of stirring [1]. In recent decades, numerous industrial applications of this phenomenon have been successfully developed. From large scale mixers and electrical arc furnaces [2,3] up to crystal growth techniques [4] electromagnetic stirring can enhance energy efficiency and improve casted material quality [5,6].

One promising application of this technology is EM stirring during solidification. The control of convective flows during crystallization allows one to influence the micro and macro structure of solidifying melts [7–9]. Previously, using the method of neutron radiography, it was shown how a traveling magnetic field affects the shape of a liquid/solid interface [10]. Moreover, the effect of the travelling magnetic field (TMF) pulsation frequency on the solid/liquid interface curvature is investigated. However, to fully understand the interaction of the TMF inductor parameters and the dynamics of the solidification process, numerical studies supplementing the experiment are necessary.

To do this, it is proposed to create a numerical model of the process under consideration, described in the first part of this article.

The second part of the work is devoted to studying the parameters of hydrodynamic flows under the influence of pulsating TMF. As is known, stirring efficiency and consequently efficiency of the technological process depend on turbulent kinetic energy. Recently there was a tendency to use pulsating, reversing or modulated magnetic fields. These kinds of unstationary applied magnetic fields allow one to increase the turbulent kinetic energy without any additional power consumption or reducing it in case of a pulsating field [11].

One of the studies in this field was completed by Eckert et al. [12], and the main idea was to change the direction of rotating magnetic field by periodic signal. The authors note that an unstationary EM field generates an unstable melt flow structure, which allows one to avoid the formation of segregations in the axial area of the ingot. Further experimental investigation was carried out by Wang et al. [13] to determine the travelling magnetic field's influence on the GaInSt melt flow, which proved the efficiency of the reversed TMF method by means of ultrasound Doppler velocimetry. The authors derived the equation for determining the optimum modulation frequency valid for this partial case.

Hachani et al. [14] have showed how the modulated TMF can affect the solidified alloys. The experimental study confirms that EM forced convection in these circumstances promotes equiaxed grain structures, and effectively reduces the formation of macrosegregations and development of segregated channels. The results showed a way to improve the quality of solidified metals and make the area of research relevant at the moment. However, because liquid metals are completely opaque, experimental investigation cannot show the full picture of the electromagnetic stirring process. Consequently, we do not have a full explanation of interaction between the modulated TMF and convective flows. A numerical simulation is a very attractive tool for understanding processes inside of opaque melt flow. The first attempt at numerical analysis of melt flow under reversed TMF was made by Oborin et al. [15]. Simulations based on the Reynolds Averaged Navier-Stokes (RANS) turbulence modelling approach were carried out for a 2D case. The results demonstrated that the travelling magnetic field, whose direction was periodically reversed, increases the stirring efficiency by several times.

On the other hand, Ben-David et al. [16,17] and later Avnaim et al. [18,19] have developed a 3D numerical model taking into account the solidification process. The influence of EM forced convection is clearly shown. But unfortunately the calculations of modulated travelling magnetic influence on solidification front shape and dynamics are still not carried out.

Thus, a review of the literature showed that in the scientific community there is a strong opinion that this technique still has big potential for increasing efficiency for several applications. And in particular, the definition of the pulsation frequency influence on hydrodynamic flow parameters is proposed to be studied both numerically and experimentally in the second part of this article.

A more detailed overview of recent studies on a modulation magnetic field liquid metal processing is summarized in the Appendix A Tables A1 and A2.

## 2. Methods

As mentioned in the introduction, numerical simulation was chosen as the main tool for this study. Simulation of the solidification process with electromagnetic effects can be nominally divided into 3 parts: Electromagnetic, Hydrodynamic and Solidification.

### 2.1. Electromagnetic Part

2.1.1. Governing Equations

The electromagnetic part was calculated in ANSYS Mechanical APDL, based on Maxwell equations:

$$\nabla \times \{H\} = \{J\} + \left\{ \frac{\partial D}{\partial t} \right\} = \{J_s\} + \{J_e\} + \{J_v\} + \left\{ \frac{\partial D}{\partial t} \right\} \tag{1}$$

$$\nabla \times \{E\} = -\left\{ \frac{\partial B}{\partial t} \right\} \tag{2}$$

$$\nabla \cdot \{B\} = 0 \tag{3}$$

$$\nabla \cdot \{D\} = \rho_e, \tag{4}$$

where $\nabla\times$—curl operator; $\nabla\cdot$—divergence operator; $\{H\}$—magnetic field intensity vector; $\{J\}$—total current density vector; $J_s$—applied source current density vector; $J_e$—induced source current density vector; $J_v$—velocity current density vector; $D$—electric flux density vector; $t$—time; $E$—Electric field intensity vector; $B$—magnetic flux density vector; $\rho_e$—electric charge density.

The continuity equation follows from taking the divergence of both sides of (1):

$$\nabla \cdot \left[ \{J\} + \left\{ \frac{\partial D}{\partial t} \right\} \right] = 0. \tag{5}$$

All of the calculations parts are performed in three dimensions formulation by edge-based magnetic vector potential method. The differential equations governing that method are following:

$$\nabla \times [[\mu] - 1]\nabla \times \{A\} + [\sigma]\left( \left\{ \frac{\partial A}{\partial t} \right\} + \nabla V_e \right) + [\varepsilon_p]\left( \left\{ \frac{\partial^2 A}{\partial t^2} \right\} + \nabla \left\{ \frac{\partial V_e}{\partial t} \right\} \right) = 0 \tag{6}$$

$$\nabla \cdot \left( [\sigma]\left( \left\{ \frac{\partial A}{\partial t} \right\} + \nabla V_e \right) + [\varepsilon_p]\left( \left\{ \frac{\partial^2 A}{\partial t^2} \right\} + \nabla \left\{ \frac{\partial V_e}{\partial t} \right\} \right) \right) = 0 \tag{7}$$

$$\nabla \times [[\mu] - 1]\nabla \times \{A\} = \{J_s\}, \tag{8}$$

where $[\mu] = \mu_0\mu_r$—magnetic permeability matrix, $\mu_0$ is the permeability of free space and $\mu_r$ is the relative permeability; $\{A\}$—Magnetic vector potential; $V_e$—electric scalar potential; $[\varepsilon_p]$—permittivity matrix.

Induced current density in the liquid metal domain is defined as:

$$\{J\} = [\sigma][\{E\} + \{v\} \cdot \{B\}], \tag{9}$$

where $[\sigma]$—electric conductivity matrix; $\{v\}$—velocity vector.

Also, we introduce the assumption of non-induction formulation and neglect magnetic field advection by liquid metal velocity due to low Reynolds magnetic number ($Re_m = \mu_0\sigma vL = 4\pi10^{-7} \cdot 3.86 \cdot 10^6 \cdot 0.030 \cdot 0.12 = 0.0756 \ll 1$, where $v$ is a typical velocity and $L$ is a typical length scale of the flow), then the expression (9) takes the form:

$$\{J\} = [\sigma]\{E\}. \tag{10}$$

In harmonic analysis, the time dependence of variable can be described by the following periodic function:

$$A(t) = A \cdot \cos(\omega t + \phi), \tag{11}$$

where *A*—Amplitude (peak) of variable; $\omega$—angular frequency; $\phi$—phase angle.

Harmonic Lorentz force:

$$F(t) = J(t)B(t) \tag{12}$$

and the time average Lorenz force density value is:

$$\{F\} = \frac{1}{2}(J_r B_r + J_i B_i), \tag{13}$$

where subscripts $_i$ and $_r$ denote imaginary and real part of the complex value.

Joule heat power density can be obtained as:

$$Q = Re\left(\frac{1}{2n}\sum_{i=1}^{n_{points}} \rho_r\{J\} \cdot \{J\}^*\right), \tag{14}$$

$n_{points}$—the number of integrating points; $\rho_r$—electrical resistivity.

The current density in the coil is given by:

$$J_{coil} = I \cdot n_{coil}, \tag{15}$$

where *I*—supplied current; $n_{coil}$—the number of coil turns. In this case, the induced current density in the coils is not taken into account.

### 2.1.2. Boundary Conditions and Numerical Mesh

As a boundary condition, the magnetic vector potential is set as zero on all outer areas of the air domain.

The numerical mesh for electromagnetic analysis is shown in Figure 1. Built up from tetrahedral elements with ten nodes of the SOLID98 type (see the Mechanical APDL Theory Reference. APDL Mechanical Element Library http://www.mm.bme.hu/~gyebro/files/ans_help_v182/ans_elem/Hlp_E_SOLID98.html). The size of the element in the metal domain was refined towards the inductors due to the skin effect.

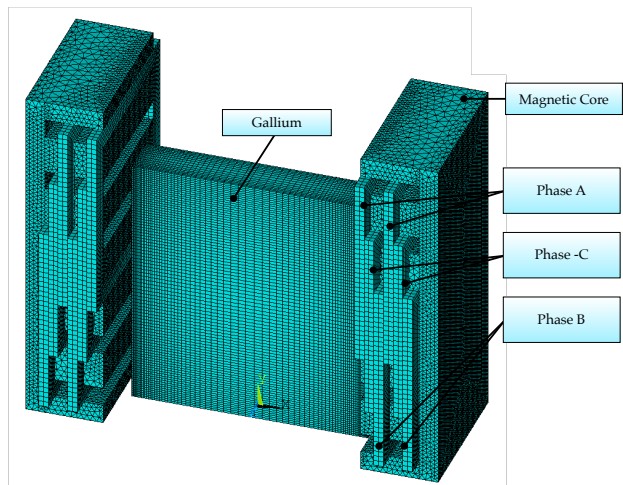

**Figure 1.** Numerical model geometry and finite element mesh for electromagnetic computation.

### 2.2. Hydrodynamic Part

Liquid metal is considered incompressible and single flow motion is described by the continuity equation

$$\frac{\partial \rho}{\partial t} + \nabla \cdot (\rho \vec{V}) = 0 \tag{16}$$

and momentum conservation equation:

$$\frac{\partial}{\partial t}(\rho \vec{V}) + \nabla \cdot (\rho \vec{V} \vec{V}) = -\nabla p + \nabla \cdot (\overline{\tau}) + \rho \vec{g} + \vec{F},$$ (17)

where $p$ is the static pressure, $\overline{\tau}$ is the stress tensor, and $\rho \vec{g}$ and $\vec{F}$ is gravity and external body force. The external body force in our case is an electromagnetically induced Lorentz force from Equation (12).

As boundary conditions the upper surface was set as slip and all remaining as no-slip. To simplify the model we assume that free surface fluctuations and deformations by fluid flow are neglected. The upper surface is a flat area.

In most electromagnetic stirring applications a turbulent or transitional from laminar to turbulent modes are present [1]. In the case considered, the flow under investigation presents a Reynolds number around $10^5$. Taking into account turbulence, these equations was calculated in two ways. The way to take turbulence into account in both cases is presented below.

### 2.2.1. RANS Modeling

Due to the fact that crystallization takes a rather long time compared with normal CFD practice, it is necessary to use a simple but reliable approach for modelling turbulent flows [20]. And for the first part of this work—solidification, we use realizable $k - \varepsilon$ model. This kind of two-equation ($k$—turbulent kinetic energy; $\varepsilon$—turbulent dissipation) Reynolds averaged model can provide a good result of mean flow velocity and does not require a lot of computational power.

Reynolds Averaging means that velocity term in Navier-Stokes equations is decomposed into the mean $\overline{u}_i$ and fluctuating $u'_i$ components: $u_i = \overline{u}_i + u'_i$.

Further, substituting expressions of this form for the flow variables into the instantaneous continuity and momentum equations and taking time average yields to the ensemble-averaged momentum equations with additional Reynolds stresses term $-\rho \overline{u_i u'_i}$ where overline means time-averaged. To find the Reynolds stresses, the Boussinesq hypothesis is used:

$$-\rho \overline{u_i u'_i} = (\frac{\partial u_i}{\partial x_j} + \frac{\partial u_j}{\partial x_i}) - \frac{2}{3}(\rho k + \mu_t \frac{\partial u_k}{\partial x_k})$$ (18)

This model solves two transport equations and models the Reynolds Stresses using the Eddy viscosity approach. Additional equations for $k$ and $\varepsilon$ are solved and then the turbulent viscosity $\mu_t$, is computed as follows:

$$\mu_t = \rho C_\mu \frac{k^2}{\varepsilon},$$ (19)

where $C_\mu = \frac{1}{A_0 + A_s \frac{kU^*}{\varepsilon}}$ is differs from standard $k - \varepsilon$ model (it is not constant). $A_0$ and $A_s$ are the constants and $S^*$ is a function of the mean strain and rotation rates.

However, this approach is used for the first part of work: steady applied TMF impact on the solidification process and for pulsed MF cases we used more precise LES approach.

### 2.2.2. LES Approach

For the second part of the paper describing flow behaviour in pulsating magnetic fields, we used the large eddy simulation (LES) approach. In LES, large eddies are resolved directly, while small eddies are modelled. Consequently, large eddy simulation occurs between DNS and RANS. The computational cost involved with LES is normally orders of magnitude higher than that for steady RANS calculations. To calculate subgrid-scale stresses we used the simple Smagorinsky-Lilly model.

Based on previous works on electromagnetic stirring ( Appendix A.1), we built the Finite volume mesh consists of $90 \times 84 \times 15 = 113,400$ elements. The size distribution of the elements is made in such a way that along the $x$ and $y$ directions in the near-wall regions, the element size decreases. The $y^+$ parameter for our case is equal to 1...0.05, so the mesh element size was $\sim$0.3 mm and $\sim$6 mm in a

central area of the volume. In the $z$ direction, the distribution of the elements was uniform with an element size of 1.3 mm.

## 2.3. Solidification

Solidification of pure gallium was solved by means of the well known enthalpy-porosity method [21] taking into account thermal buoyancy. In this method, the enthalpy of the material is calculated as the sum of the latent heat $h$ and sensible enthalpy $\Delta H$:

$$H = h + \Delta H. \tag{20}$$

The energy equation for solidification:

$$\frac{\partial}{\partial t}(\rho H) + \nabla \cdot (\rho \vec{v} H) = \nabla \cdot (k \nabla T) + S \tag{21}$$

where $S$ is a source term.

The mushy zone in enthalpy-porosity method is treated as a porous medium and momentum equation written as:

$$S = \frac{(1-\beta)^2}{\beta^3 + 0.001} A_{mush} \vec{v} \tag{22}$$

where $\beta$ is liquid fraction, $A_{mush} = 10^5$ is the mushy zone constant

The boundary conditions are as follows. On the lower surface, a constant temperature of 15 °C is set, lateral walls are considered as adiabatic and on the upper surface, an outgoing heat flux of 0.5 W/m$^2$ is set.

The computations were performed for a rectangular container of $0.12 \times 0.10 \times 0.02$ m filled with liquid Gallium and the thermophysical properties used are listed in Table 1.

**Table 1.** Gallium thermophysical properties [19,22].

|  | | Property | Value | Units |
|---|---|---|---|---|
| Ga (solid) $298 < T[K] < 303$ | $\rho$ | density | 5910 | kg/m$^3$ |
| | $k$ | thermal conductivity | $60.66 - 0.183 \cdot T$ | W/m ·K |
| | $C_p$ | specific heat | | 396 J/kg · K |
| Ga (liquid) $303 < T[K]$ | $T_s$ | Solidification temperature | 303 | K |
| | $\rho$ | density | $6262 - 0.000612 \cdot T$ | kg/m$^3$ |
| | $k$ | thermal conductivity | $15.7 + 0.031 \cdot T$ | W/m ·K |
| | $C_p$ | specific heat | 407 | J/kg ·K |
| | $\mu$ | dynamic viscosity | $0.0156 - 1.053 \cdot T$ | kg/m ·s |
| | $\Delta H$ | latent heat | 80200 | J/kg· K |
| | $\sigma$ | electrical conductivity | $3.86 \cdot 10^6$ | S/m |

Fluid dynamic and solidification process simulation were performed by means of Finite volume method based ANSYS FLUENT 16.0 commercial software (ANSYS, Inc.).

## 2.4. Simulation Coupling

A schematic sketch of the described numerical model is shown in Figure 2. Resulting EM force is transferred to the HD part, then the flow characteristics determine the solidification circumstances and vice versa.

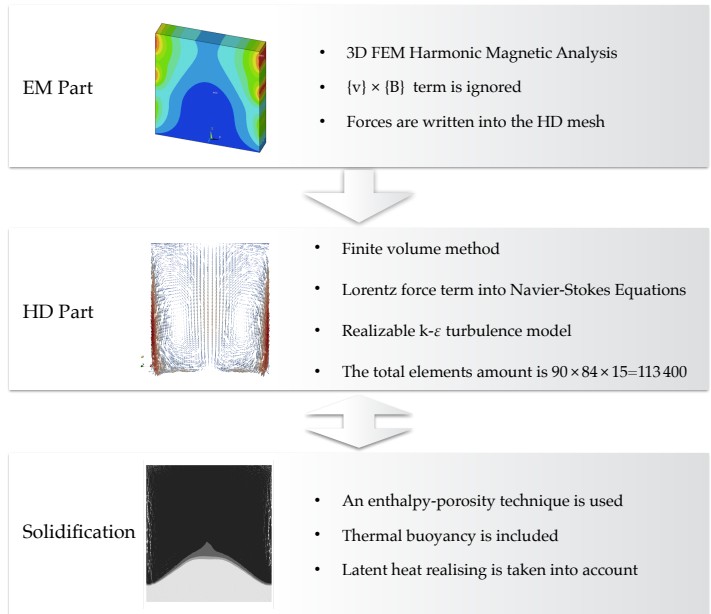

**Figure 2.** Numerical scheme for the solidification process in presence of travelling magnetic field (TMF).

### 2.4.1. Experimental Setup and Validation

The considered TMF inductor consists of a magnetic core with six windings laid in the magnetic core slots and is shown in Figure 1. These windings are connected to a three-phase power inverter Mitsubishi Electric Inverter D-700 SC. The frequency is 100 Hz. Coils connected in such way that they generate travelling magnetic field that is, A(-C)B(-A)C-B or AZBXCY, which means that phase shift between windings in nearby slots is equal to 60° and the number of slots per pole and phase is 1. Thus, they generate a travelling magnetic field directed downward and the wave length is 0.162 m. The number of turns in the windings $n$ is equal to 120.

The magnetic cores are made from laminated electrical steel and have a relative magnetic permeability equal to 30. The magnetic field around the coils closes along the teeth of the magnetic core. Then, through an 5 mm air gap, the magnetic field penetrates the liquid metal.

In an electrically conductive metal, an electromotive force and eddy currents occur under the action of an alternating magnetic flux. The interaction of these currents with the magnetic field of the inductor produces Lorentz forces in the liquid metal. These forces are the source term in the equations describing hydrodynamic flow.

The vessel of liquid gallium is placed between two TMF inductors (Figure 3). Crystallization is obtained by applying a temperature of 15 °C to the lower surface of the metal using Peltier elements. The following measurements were made on this installation.

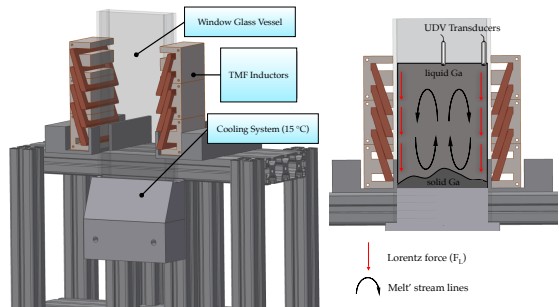

**Figure 3.** Experimental setup of solidification process in presence of travelling magnetic field from Reference [10].

### 2.4.2. UDV Measurements

To study the liquid gallium flow under electromagnetic influence, the well-known method of ultrasound Doppler velocimetry (UDV) was used. UDV transducers were located on the upper surface near the lateral wall region and on the central axis of the vessel, as shown in Figure 3. The direction of the UDV ray was aligned with the $y$ axis, and the $y$-component of velocity was measured. As a velocimeter the DOP 3000 of Signal Processing was used.

### 2.4.3. Neutron Radiography Visualisation

The experimental data on the dynamic position and shape of the liquid/solid interface were obtained previously at the thermal neutron transmission radiography facility, NEUTRA (Paul Scherrer Institute, Villigen, Switzerland) , and are presented in Reference [10]. The neutron radiography method is a promising and unique measuring technique. The principle of operation and experimental setup are described in detail in Reference [10]. In short, the principle of operation means the neutron beam is directed at the crystallizing alloy and neutron transmission depends on a material density. Then, this neutron beam passing through the material is converted into light emission and is an indicator of the phase transition.

## 3. Results

As a result of coil current the magnetic field around coil will appear. In Figure 4a spatial distribution of magnetic flux density is shown. The main magnetic fluxes are concentrated in the magnetic core of inductors. The active part of the current is directed along or against axis $z$. Consequently the magnetic flux goes around the $z$ axis through the teeth and back parts of magnetic core. Then the magnetic flux closes through air gap and liquid metal.

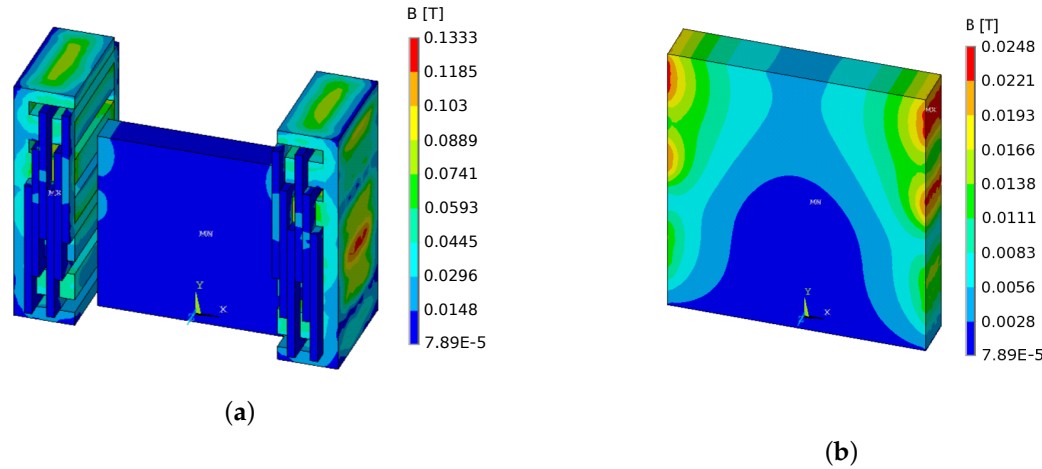

(**a**)

(**b**)

**Figure 4.** Magnetic flux density on surfaces of whole TMF stirrer (**a**) and melt volume (**b**).

Figure 4b shows the magnetic flux density on a surface of liquid metal volume. The maximum value reaches 25 mT. The field intensity decreases close to the axis of symmetry. In addition, it should be noted that we consider the simulation result for zero time, this means that the first coil current (phase A) is maximum, and the remaining two are less by $\sqrt{2}$ times.

The effect of an alternating magnetic flux on an electrically conductive media, as known, generates electromagnetic forces or Lorentz forces within it. Figure 5 shows the distribution of time averaged Lorentz forces in a liquid metal. Due to the fact that the magnetic field and eddy currents are mainly concentrated on the lateral surfaces, Lorentz forces also act in these zones. On the axis of vessel symmetry, their magnitude exponentially decays. The direction of forces coincides with the

direction of TMF–downward. This distribution of force vectors is the source term in the Navier-Stokes Equation (17).

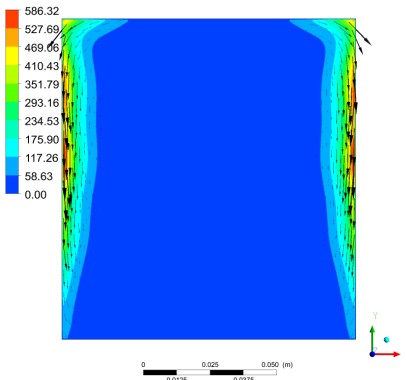

**Figure 5.** Cut surface of liquid metal volume with Lorentz forces [N/m$^3$].

As expected, under such Lorentz force action we get a two-vortex flow pattern as shown in Figure 6a. In the near-wall areas, with an increased concentration of force, the liquid metal develops a maximum velocity of about 30 mm/s. The liquid then rises along the central axis already with lower intensity.

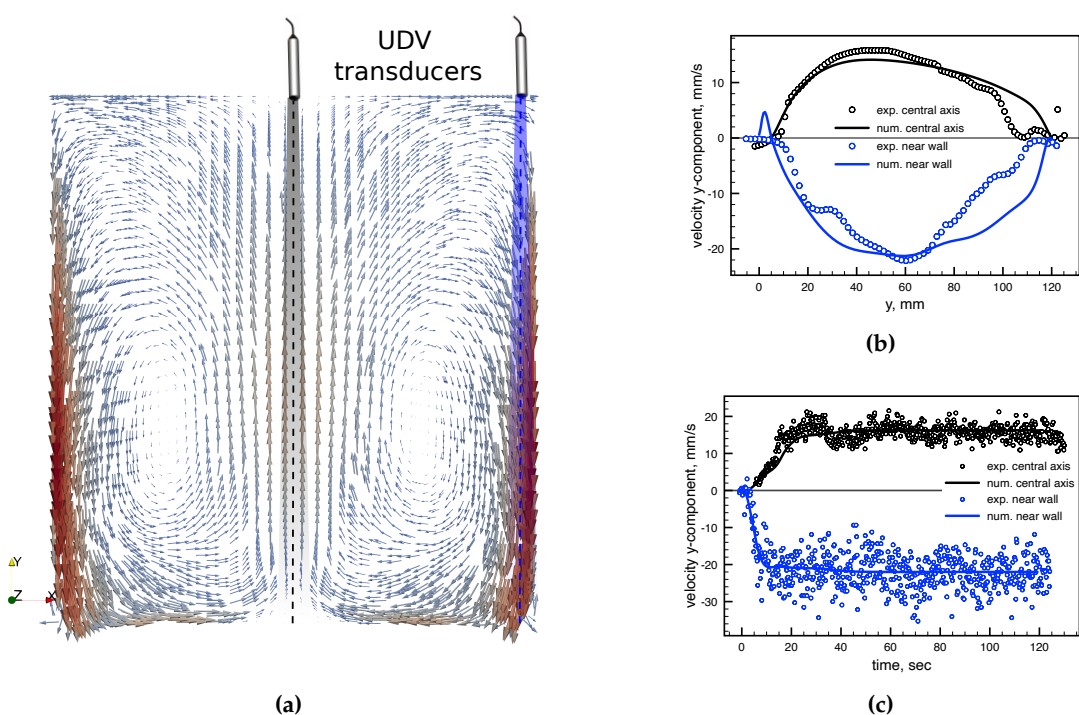

**Figure 6.** (**a**)—velocity field of liquid metal under TMF impact with ultrasound Doppler velocimetry (UDV) transducers placement. (**b**)—mean velocity profile, (**c**)—velocity dynamics in points at the melt height of 45 mm. Blue color corresponds to near the wall area, black—central axis, whereas dots are experimental data and lines—numerical data.

Verification was performed using two UDV transducers. And Figure 6b shows a comparison of numerical (line) and experimental (dots) mean velocity profiles along the *y* axis. We can note a good agreement of the results. Velocity dynamics were also analyzed. As we can see from the Figure 6c,

the $k - \varepsilon$ turbulence model does not capture all the pulsations of the fluid velocity, but it determines the integral value well.

### 3.1. Solidification

Furthermore, we have simulated the solidification process under such fluid flow conditions. Figure 7 shows snapshots of solidification dynamics. The main heat flux must go through the lower surface and therefore the solidification starts from the bottom. Then, warmer volumes of metal, with the help of TMF, descend in the areas of the side walls. This circumstance slows down the rate of crystallization. On the other hand, in the central region, the flows of chilled metal are directed upward, which accelerates crystallization in this region. In the same way, we have a curvature of liquid/solid interface as shown. The lower line of images is an experimental data. From blue to green corresponds to already solidified metal and from red to yellow is a melt. We should note that the results of numerical modelling are in good qualitative agreement with experimental images and we can conclude that this model is a good tool and can be used for further investigations.

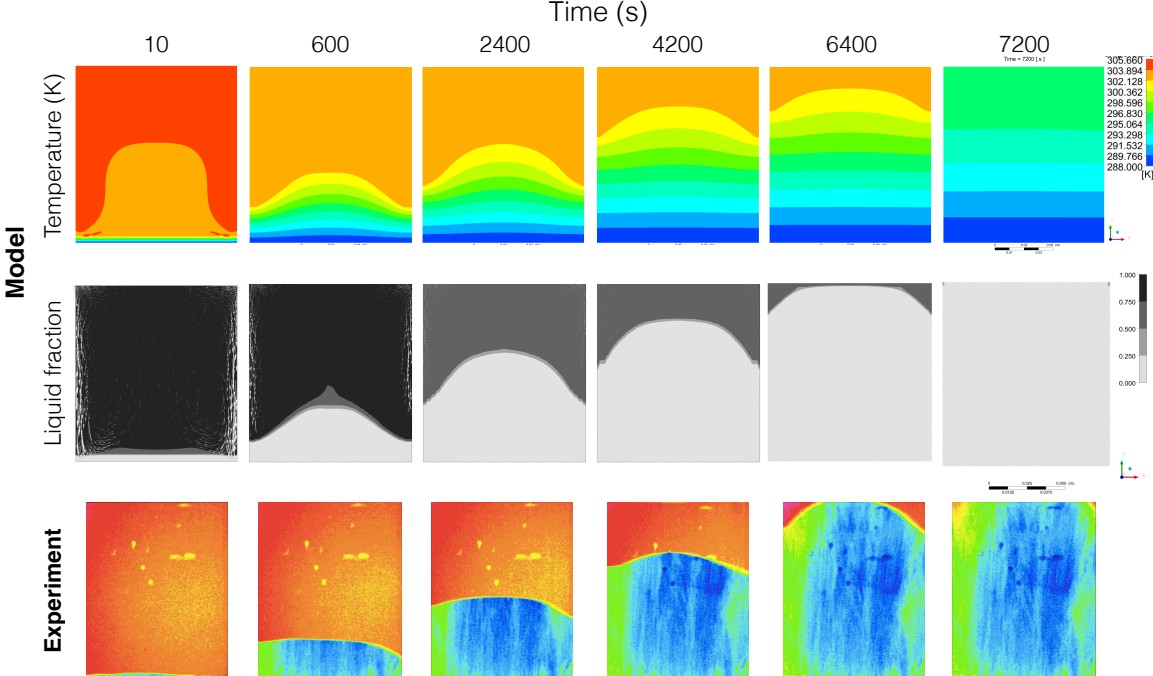

**Figure 7.** The solidification process dynamics in the presence of a travelling magnetic field. Velocity vectors are superimposed to liquid fraction maps (white arrows). The animation of transient solidification can be found at supplementary data http://dx.doi.org/10.17632/7fmy5kh278.1.

Another interesting point is the change in velocity during solidification. If we a look at the liquid fraction in Figure 7 at the beginning of solidification the velocity vectors are large. Then, after 2400 s, velocity vectors almost disappear. To analyze this phenomenon the comparison of velocity profiles at different time steps is shown in Figure 8. These profiles are taken from the near-wall area according to Figure 6b. As we can see, decrease starts after 600 s and when half the volume is solidified the velocity damped by 3 times. This phenomenon can be explained by mushy zone transport into liquid metal bulk. But there is a decreasing of EM forcing parameter during solidification. It may affect the resulting heat flux in a liquid volume. Avnaim et al. [19] carried out such investigation and suggest increasing the EM forcing parameter during solidification to keep convection condition constant.

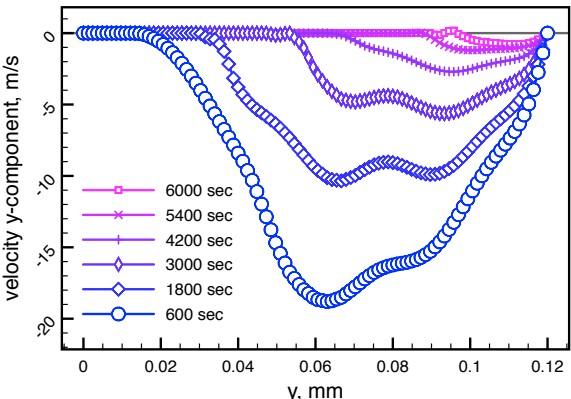

**Figure 8.** Melt velocity decreasing during solidification. The location of line is correspond to the near wall UDV transducer.

### *3.2. Pulsed TMF Study*

The next part is devoted to pulsed TMF action on a gallium melt. For this part we used the same experimental setup, but without solidification. The magnetic field was periodically interrupted with frequencies $f_p = 1; 0.5; 0.3.$ and $0.1$ Hz. The main difference in the numerical model is a LES turbulence model instead of RANS. This approach gave a more proper calculation of fluid fluctuations.

### 3.2.1. Turbulence Model Testing

First of all we tested the turbulence models in a pulsation mode. Results of velocity at $f_p = 0.5$ Hz can be seen in a Figure 9. The grey columns represent pauses and the white ones are active half periods. We can note that all the compared models show good convergence, but the LES model in some cases slightly differs from the RANS models. Moreover, the LES approach is more suitable for predicting low-frequency velocity fluctuations in two vortex flow patterns [23]. Therefore, it was decided to use this approach.

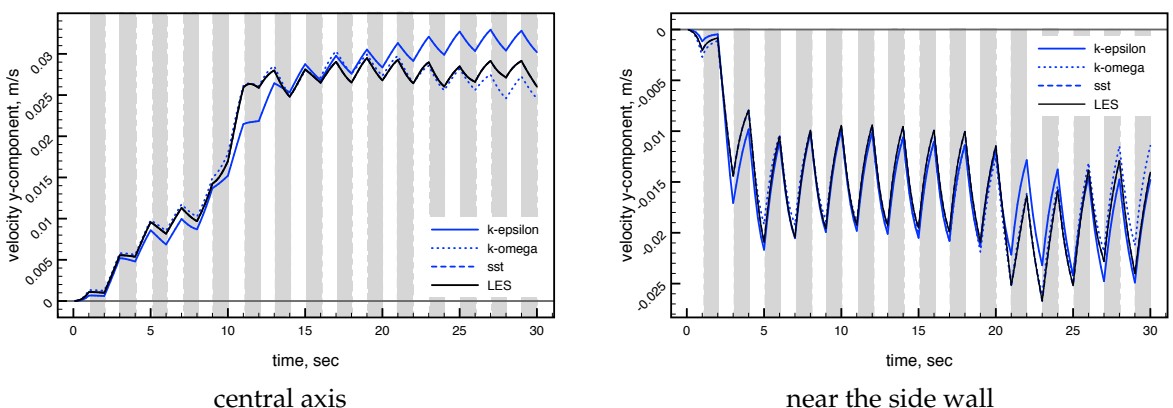

central axis          near the side wall

**Figure 9.** Turbulence models testing.

### 3.2.2. Spin-up Behaviour

We have investigated spin-up behaviour of liquid metal under pulsed TMF in both numerically and experimentally. Results for velocity at different pulsation frequencies $f_p$ are shown in Figure 10. The positive velocities are taken in the central area and the negative data correspond to the near-wall area. At first glance all cases look similar. For the velocity at the central area experimental data are similar. We cannot conclude that in the central region we observe pronounced velocity pulsations, both in numerical data and in experimental data. But if we look at more intense flows at the near-wall area, we will clearly see periodic ripples. For numerical data, pronounced periodic fluctuations

corresponding to the pulsation frequency are observed. Experimental data, as it seems, does not always show these fluctuations. But it quantitatively supports numerical data well.

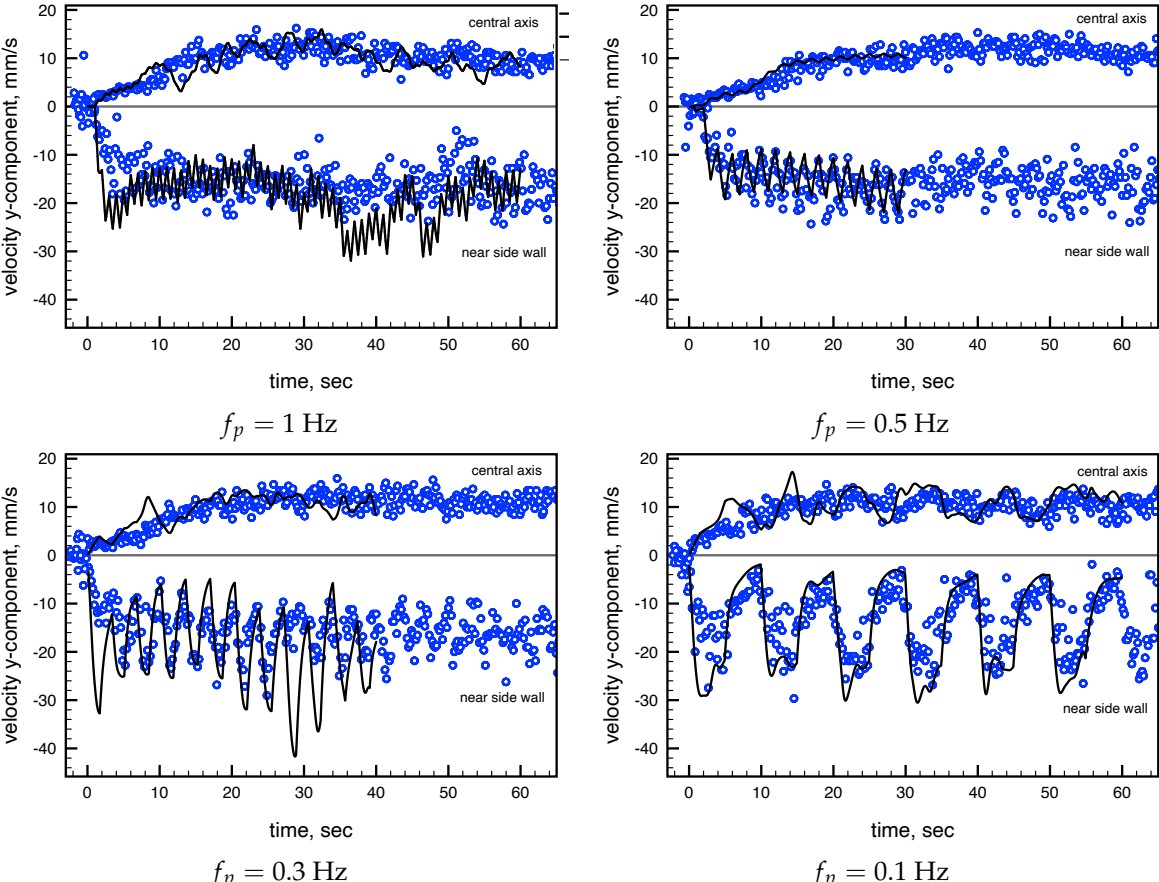

**Figure 10.** Spin-up liquid metal velocities at different pulsation frequencies. Dots–measured; lines–calculated. Video files of simulations can be found at supplementary data http://dx.doi.org/10. 17632/7fmy5kh278.1.

Spin-up time for the near-wall area reaches 2–4 s, while in the central area flow needs 10–20 s to reach working velocity. The pulsation frequency almost does not affect spin-up time. Fluctuations after spin-up time can be explained by the rectangular shape of vessel [24,25].

### 3.2.3. Pulsation Frequencies Analysis

To analyze the influence of pulsation frequency on a liquid metal flow a representative comparison is given in Figure 11. Numerical results remain unchanged, but experimental data was processed by FFT (Fast Fourier Transformation) Smooth. This function uses as effective smooth width the value from the Factor (=2) field. Such data processing can help to reduce "noise" and to see more clearly the experimentally obtained pulsation frequency.

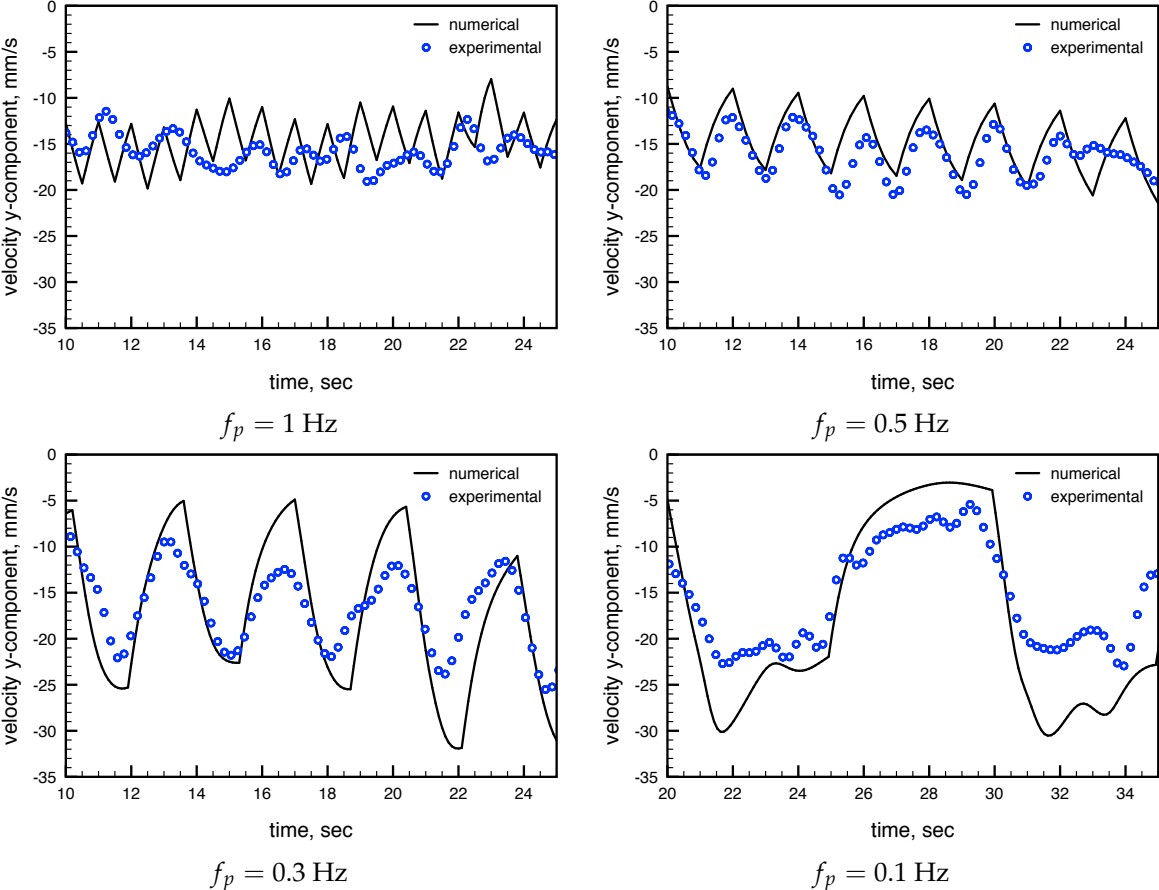

**Figure 11.** Comparison of experimentally (after FFT smooth) and numerically obtained velocities at different pulsation frequencies in near wall region. Animations of transient calculations can be found at supplementary data http://dx.doi.org/10.17632/7fmy5kh278.1.

First of all, it is worth noting from the numerical results that the magnetic field pulsation frequency is equal to the frequency of the liquid metal flow pulsations. At high frequencies (1 and 0.5 Hz), the flux does not have time to develop a large value of maximum velocity because a pause in the second half-period of magnetic field function begins and the flow decays. On the other hand, at a low frequency (0.1 Hz), the first (active) half-period of the modulation periodic function of the magnetic field lasts 5 s. During this time, the flow manages to reach the maximum value of velocity. However, this is not an optimal mode, since the flow velocity begins to "saturate". Moreover, at this frequency, when the flow developed, some slight oscillations are present. These "secondary" fluctuations may be the result of a square shape of the vessel, azimuthal flow or interaction of two vortices and need to be studied more closely. The case for $f_p = 0.3$ Hz is the closest to the optimal one, according to Reference [26], because the force action interruption is carried out on the knee of the curve.

Numerical and measured data have good agreement in the last three cases: 0.5; 0.3 and 0.1 Hz. The amplitude of measured value on graphs is less than the numerical one, because of FFT smoothing. Experimental data even shows some small fluctuations in the case of $f_p = 0.1$ Hz. However, for case UDV measurements does not catch flow pulsation properly. For such frequency and higher, another approach should be used, for example potential probe technique.

## 4. Discussion

To evaluate the results obtained in the second part of this paper: pulsed flow analysis, it is necessary to compare it with the results of other works. If we look at previous studies of a time-modulated magnetic field, we can see that the authors suggest a variety of optimal modulation

frequency criteria (see Appendix A.2). The most suitable for our case works [10,26] suggests EM field modulation frequencies

$$f_m = \frac{1}{2t_s} \tag{23}$$

where saturation time is $t_s = \frac{a^2}{v_t}$, $a$—half width of the cell in $y$ direction, $v_t$—eddy viscosity.

And

$$f_{ch} = \frac{1}{T_{ch}} \tag{24}$$

where $f_{ch}$—characteristic frequency; $T_{ch} = \frac{2\pi \frac{r_r + r_z}{2}}{V_{ch}}$; $r_r$ and $r_z$—radial and axial dimensions of the vortex.

In recalculating the case considered in this article, these frequencies will be $f_m = 0.04$ Hz and $f_{ch} = 0.321$ Hz respectively.

To analyze the influence of TMF pulsation on flow parameters in Figure 12 results of simulations and measurements are summarized. The amplitude of pulsations, as expected, strongly depends on pulsation frequency for both numerical and experimental cases. But for maximum value we do not have a linear dependence. An extremum explicit for numerical one exists at a pulsation frequency of 0.3 Hz. However, this extremum was not obtained experimentally and almost flat area of curve between 0.1 and 0.3 Hz is shown. These results correlate well with so-called "MHD resonance" value mentioned in References [10,11]. Moreover, applying a pulsating function to EM fields means that we will have interruptions and, constituently, consume less electrical energy. If time of acting half period is equal to acting one, the energy consumption will be less in two times. Then, the data of maximum velocity in steady applied case (0 Hz) and for pulsating cases differ not so dramatically.

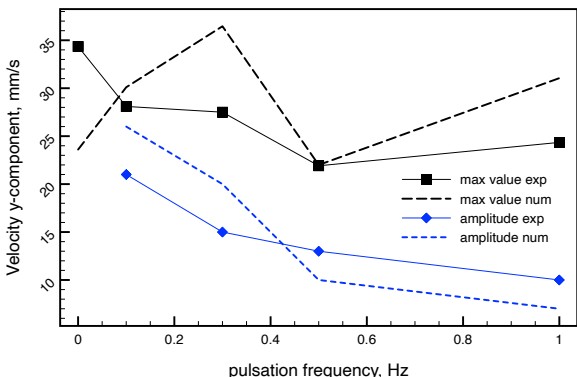

**Figure 12.** Dependencies of liquid metal flow parameters on TMF pulsation frequency.

The next step should be a numerical study of solidification under modulated TMF. That simulation will provide a better understanding of the interaction mechanism between non-stationary EM forced convection and liquid solid interface.

## 5. Conclusions

The numerical model of solidification of pure Ga process in the presence of TMF is developed. This model is verified by comparison with velocities measurements and transient liquid/solid interface position snapshots. A developed model can be used for further investigations.

Analysis of pulsed TMF action on liquid metal flow is carried out. An investigation is done both numerically and experimentally and shows good agreement for pulsation frequencies from 0.5 to 0.1 Hz. The dependencies of maximum value of velocity and amplitude on pulsation frequency are obtained. The extremum for numerical maximum velocity is 0.3 Hz and it is correlated with other works. The pulsed EM liquid metals processing can reduce the electrical energy consumption with

minimal loss in efficiency. This approach can de used not only for solidification processing but also for numerous metallurgical processes.

A literature review of liquid metal processing by time modulated EM fields research area is completed. This review shows the prospects of such an approach and issues not yet investigated.

Further investigations should be devoted to the simulation of solidification process under time-modulated TMF action and to obtaining dependencies of liquid/solid interface curvature Another interesting point is investigation macro segregation (freckles, channels etc.) under described pulsed flow action and columnar to equated transition at this external impact.

**Supplementary Materials:** Supplementary data associated with this article can be found on http://dx.doi.org/10.17632/fnb6drkf93.1 (is the UDV measurements data) and http://dx.doi.org/10.17632/7fmy5kh278.1 (video files of transient numerical simulation).

**Author Contributions:** EM numerical model D.K. and E.S.; HD model and investigation E.S.; review, editing, and supervision E.B. and D.K.; methodology D.K., writing—original draft preparation E.S., solidification experiment D.K., project administration E.B. All authors have read and agreed to the published version of the manuscript.

**Funding:** The work was supported by the Russian President scholarship for study abroad.

**Acknowledgments:** Evgeniy Shvydkiy acknowledges Tristan Fehling for his assistance in CFD simulation and Francis Scarr for manuscript proof reading.

**Conflicts of Interest:** The authors declare no conflict of interest.

## Abbreviations

The following abbreviations are used in this manuscript:

| | |
|---|---|
| AMF | Alternating Magnetic Field |
| DOF | Degree of Freedom |
| EM | Electromagnetic |
| HD | Hydrodynamic |
| HMF | Helical Magnetic Field |
| LES | Large Eddy Simulation |
| MF | Magnetic Field |
| MHD | Magnetohydrodynamics |
| RANS | Reynolds-averaged Navier–Stokes |
| RMF | Rotating Magnetic Field |
| TMF | Travelling Magnetic Field |

## Appendix A. Review on a Previous EM Stirring Studies

*Appendix A.1. Previous EM Stirring of Melt in a Rectangular Cavity Investigations*

**Table A1.** Previous EM stirring of melt in a rectangular cavity investigations.

| Author | Turbulence Model | Solidification, Method | Software | HD Mesh | Material and Size |
|---|---|---|---|---|---|
| Dadzis [27] | No turbulence model is used in flow calculation in the present study. Re = 1400 | | GetDP, Elmer (thermal), OpenFOAM | A hexahedral grid with $42 \times 42 \times 27 = 47,628$ elements | Ga $100 \times 100 \times 75$ mm$^3$ |
| Kolesnichenko et al. [28–30] | $k - \omega$ SST | No | COMSOL (FEM) | 76,554 Hexahedral elements | GaSnZn $450 \times 20 \times 75$ mm$^3$ |
| Ben David et al. [16,17,31] | laminar flow | Yes, C. Prakash, V.R. Voller approach | COMSOL | 10,469 elements with 60,000 DOF and 32,144 with 176 610 DOF [31] [16], 80,485 and the number of DOF was 73,550 for [17] | Ga (and GaInSn [17]) in a rectangular container of $0.06 \times 0.06 \times 0.09$ m$^3$. |
| Avnaim et al. [18,19,32] | The maximum Reynolds number is not greater than 2500. 3D Direct Numerical Simulation (DNS) | Yes, Multi-Domain method | COMSOL Multiphysics 5.0 (FEM) | 21,500 (120,000 DOF) Lagrange's elements [18] and 50,000 (590,000 DOF) [19], 50,400 (275,110 DOF) for VOF method and 22,890 (708,420 DOF) for the MD modell [32]. | The gallium in cavity with the dimensions $0.06 \times 0.06 \times 0.09$ m$^3$. |
| Hachani et al. [14,20,33] | Realizable k-$\varepsilon$ | A three-phase volume averaged equiaxed model | ANSYS FLUENT | grid comprising $100 \times 60 \times 15$ mesh cells | Ga–In–Sn alloy $0.1 \times 0.06 \times 0.01$ m$^3$ |
| Wang et al. [13,26] | 2D Theoretical model | No | Analytical | 2D Theoretical model | GaInSn $0.1 \times 0.06 \times 0.01$ m$^3$ |
| Köppen et al. [10,34] | Exp. | Yes | Exp. | Exp. | $0.12 \times 0.11 \times 0.02$ m$^3$ (two inductors) |
| Dzelme et al. [35,36] | DES and $k - \omega$ SST | No | ANSYS Classic or Elmer and OpenFOAM | 1.035M or 300 k elements in work [36] 37,500 elements in work [35] | gallium $0.1 \times 0.1 \times 0.03$ m$^3$ |
| Oborin et al. [15] | 2D a $k - \varepsilon$ turbulence model | No | OpenFOAM for velocity field and an analytical solution for the electromagnetic body force | $500 \times 250 = 125,000$ control volumes | gallium alloy (Ga87.5%, Sn10.5%, Zn2.0%); $l_1$=24 cm, $l_2$ = 18 cm and $h$ = 9.5 cm (asymmetric cavity) |

*Appendix A.2. Previous Modulation/Pulsation of MF Studies*

**Table A2.** Previous Modulation/Pulsation of MF studies

| Author and References | Type of the MF | Modulation Frequencies | Numerical/ Experimental | Optimal Criteria |
|---|---|---|---|---|
| Eckert et al. [12,37] | RMF | 0.2; 0.475; 0.77 Hz for pulsed and 0.02; 0.08; 0.15; 0.2 Hz for reversed cases [12]. For solidification case 0.1, 0.2, 0.3, 0.35, and 0.45 Hz [37]. | Numerical (with solidification) and experimental | 0.15 Hz for max intensity of the secondary flow and 0.45 Hz for avoiding a segregation. |
| Wang X. et al. [13,26] | TMF | The investigated modulation frequencies are 6, 5, 4, 3, 2, 1, 0.5, 0.2, 0.1, and 0.05 Hz (reversed modulation) | Analytical and Experimental by UDV | The optimum modulation frequency that would allow saturation to be reached is $f_m = \frac{1}{2t_s}$ where saturation time $t_s = \frac{a^2}{v_t}$, $a$—the half-width of a cavity $y$, $v_t$—the effective viscosity. |
| Räbiger et al. [38] | RMF | Pulsed modulation with time period of ranges 2–6 sec. and 10–30 sec. | Experimental by UDV | Dependences of the secondary now intensity on the duration of the pulse cycle for different RMF intensity are found. |
| Oborin et al. [15] | TMF | Reverse frequencies are 0.08; 0.1; 1; 1.25; 10.00 Hz | Numerical in a OpenFOAM and experimental by an ultrasonic Doppler velocimeter | Introduced the coefficient of heterogeneity: $\xi(t) = \sqrt{\frac{\sum (C(\tau) - C_i(\tau))^2\vert_{\tau=t}}{(C(\tau) - C_i(\tau))^2\vert_{\tau=0}}}$ that is, through the relation between the standard deviation of the impurity concentration Ci at all n points in the cavity from the final impurity concentration after stirring (at a time t) and the same standard deviation at the initial time $t = 0$. The characteristic stirring time $t_{eff}$, which determines the time when the parameter $\xi(t)$ decreases to $\xi(t)_{eff}$ is also estimated |
| Dropka et al. [39,40] | TMF | pulsed downward TMF of 0.5 and 0.05 Hz for different modulation strength and modulation amplitude. | Global 3D numerical analysis. Flow in the melt was described by $k\omega - SST$ model by ANSYS CFX 13.0 and Ansys Classic. | The radial temperature profiles in the middle of the melt for various modulated TMF flows with unmodulated TMF driven flow are compared. The most promising flow pattern in this study was obtained for sinusoidal on-off mode of TMF pulsing with $f_p = 0.3$ Hz. Also the mass fraction dynamics are examined |
| Musaeva et al. [11] | AMF | Pulsed Lorentz force was applied in a range of the modulation frequency of 0.05 Hz $< f_p < 1$ Hz. | Numerical by the ANSYS Fluent software package using a Large Eddy Simulation (LES) turbulence model. | Pulsed AMF influence on the turbulent kinetic energy of the melt flow was investigated. To recognize the effect of the low-frequency pulsed Lorentz force on the melt mixing, a simulation of the temperature field homogenization was carried out. |
| Hachani et al. [14] | TMF | The electromagnetic force direction is periodically reversed. Electromagnetic force inversion frequency is equal to 0.125 Hz. Electromagnetic | Exp. (thermocouples; chemical method coupled with the Inductive Coupled Plasma technique and X-ray analysis) | The diffrent parameters was analayzed experimentally to estimate a TMF influence on solidification process: 1. Temperature field evolution. 2. Final metallographic structure and grain size. 3. Solute distribution. 4. Lead concentration distribution (macro- and mesosegregations) and the morphology of segregated channels. |
| Wang B. et al. [41] | HMF | reversed periodically modulation with frequencies $f_m = 0.025; 0.05; 0.1; 0.25$ Hz | Ultrasonic Doppler velocimetry (UDV, DOP 3010) was used to quantitatively measure the liquid metal flow. | The averaged axial and azimuthal velocities for various modulation frequencies with different aspect ratio of vessel are obtained. An optimal modulation frequency, under which the magnetic field could efficiently stir the solute at the solidification front, exists both in secondary and global axial flow (0.1 Hz and 0.625 Hz, respectively). |

**Table A2.** *Cont.*

| Author and References | Type of the MF | Modulation Frequencies | Numerical / Experimental | Optimal Criteria |
|---|---|---|---|---|
| Musaeva et al. [10] | TMF | Pulsed modulation of 0.1, 0.3, 0.5, 1, 2 Hz | Experimental by neutron radiography | It has been experimentally proved that a flatter shape of the solidification front and the reduced irregularity of the solid/liquid surface can be obtained with the $f_P$ value closer to the characteristic frequency of melt circulation. |
| Losev et al. [28,42] | TMF | Reversed and pulsed modulation with period of 20, 30, 40, 50 sec (0,05–0,02 Hz) | Numerical by FEM in COMSOL and experimental by UDV measurement | Average flow velocity and its root mean square vs. TMF inductor supply current and TMF reverse modulation period was analysed. The dynamics of the crystallization front position are computed by wave-length analysis. |
| Shvydkiy et al. [43,44] | TMF | Reversed and pulsed modulation | Numerical by COMSOL | A homogenization parameter of particles into the volume are implemented. Modulation is shown more effective stirring of particles in both pulsed and reversed cases. |
| Khripchenko et al. [45] | RMF and TMF | Reversed modulation with periods from 8, 16, 24 and 32 sec | Numerical by Ansys CFX and experimental | Several parameters were analyzed: Number of grains, Brinell hardness, specific kinetic energy. The analysis of the experimental results has indicated extrema on the curve, illustrating the grain size in the ingot structure, being dependent on the period of reversals of the rotating magnetic field. It is revealed that in the experimentally studied period of reverse pulsations of the rotating magnetic field the integral characteristics of hydrodynamic fields, such as the specific kinetic energy of a large-scale flow, the energy of turbulent pulsations, the kinetic energy of vertical motion, depend monotonically on the period. |

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
