# Peer review of "Liquid Metal Flow Under Traveling Magnetic Field—Solidification Simulation and Pulsating Flow Analysis"

_metals, doi:10.3390/met10040532_

Round 1

Reviewer 1 Report

Paper presents interesting results of modelling; however, following issues should be addressed before the publication:

  1. Experimental setup has to be described in more details. Please present details of cooling system as well as information about the coil’s power supply
  2. Connection between the experimental setup and mesh geometry in figure 2 is not clear. Please clearly indicate the coils location. It seems that angle of coils location is different in experiment and in model. Please explain.
  3. Please explain what solver was used for magnetic modelling. What were the convergence criteria?
  4. Turbulent flow was indicated in the section 3.2.1. Please explain what model was used for modeling the turbulent flow? It seems that there is significant uncertainty. Please explain, if it is connected with the model or with parameters assumption.
  5. Conclusions should be developed and stated in more quantitative way. Please present the practical consequences of achieved results as well as areas of further research.

Author Response

All of the corrections are added to a manuscript in blue color.

1. Experimental setup has to be described in more details. Please present details of cooling system as well as information about the coil’s power supply

Regarding cooling system: The vessel was positioned at the cooling system. The cooling system consisted of an alumina box, uniformly cooled to 15◦C by Peltier elements during the experiment to realize the conditions of Ga solidification.

Inductor windings are supplied from Mitsubishi Electric Inverter D-700 SC. The coils power supply is added to Fig. 4. Coils connected in such way that they generates travelling magnetic field i.e. A -C B -A C -B or AZBXCY, which is means that phase shift is equal 60 degrees and the number of slots per pole and phase equals 1.

2. Connection between the experimental setup and mesh geometry in figure 2 is not clear. Please clearly indicate the coils location. It seems that angle of coils location is different in experiment and in model. Please explain.

Yes, you right. The figure 2 (Experimental setup) just shows appearance of investigated installation and inductors windings are sketched. The real photo will be attached to corrected manuscript. But numerical model geometry is correspond to figure 1 (Numerical model geometry and finite element mesh for electromagnetic computation.) The more detailed description is added.

3. Please explain what solver was used for magnetic modelling. What were the convergence criteria?

For EM computation was used Jacobi Conjugate Gradient (JCG) solver (Iterative Solver) due to the fact that the model consists of a big number of elements (750052) . But we have to note that in our formulation, when saturation of magnetic core still is not taken onto account, the harmonic EM is not a problem. Convergence for EM computation was set as 10^(-5). We believe that it should be enough.

4. Turbulent flow was indicated in the section 3.2.1. Please explain what model was used for modeling the turbulent flow? It seems that there is significant uncertainty. Please explain, if it is connected with the model or with parameters assumption.

The turbulent flow was modelled in two ways: by means of Relizeble k-epsilon model in solidification case and Large Eddy Simulation (LES) approach in pulsed TMF case. For solidification case we have choose the simple put robust one because one simulation of solidification takes rather long time (100 min simulation time). And LES approach become it’s good sensitivity to flow pulsations.

5. Conclusions should be developed and stated in more quantitative way. Please present the practical consequences of achieved results as well as areas of further research.

Thank you for that remark! We have added one more conclusion that applying a pulsating function to EM fields means that we will have interruptions and constituently consume less electrical energy. If time of acting half period is equal to acting one the energy consumption will be less in two times. When the data of maximum velocity in steady applied case (0 Hz) and for pulsating cases is differ not too dramatically. And this approach can de used not only for solidification processing but also for numerous metallurgical processes.

The further research will be devoted combining solidification simulation and pulsating force action. It should give dependencies of liquid/solid interface curvature. Another interesting point is investigation macro segregation (freckles, channels etc) under pulsed flow action and columnar to equated transition at this external effect.

Reviewer 2 Report

This article provides an interesting approach to analyse a liquid metal flow and solidification under magnetic field. Experimental data are well supported by simulations and the scientific interest is not questionable. It is damaging that result analysis may be not more advanced, because it seems that some more conclusions could be drawn from what you present.

The main difficulties when reading your article are (see comments hereunder):

  1. the strange organization of the chapters
  2. some approximations and lack of explanations (in the numerical model, in the description of figures, etc.)
  3. a quite poor proof reading as far as I may note. Misprints are numerous, figures are not always located logically and their numerotation does not follow the right order, etc.

My corrections/comments/questions follow (concerning the english, I did not make an extensive corretion, so proof reading remains useful)

1: on a liquid metal
11: it was found the ability : I do not catch the sentence
13: through the manuscript, you use either "behavior" or "behaviour". Please make a choice.
19: leads to the movement of what ?
42 : the formation of segregations
46: Hachani et al. show... : you switch from past to present (see also 58)
55: Navier-Stokes
73: one could imagine a chapter 2.1.1. Equations, 2.1.2. Boundary Conditions and Numercial Mesh, 2.1.3 Experimental Setup
77: Ju : you write Jv in equation 1
80: three dimensions
81: the differential equations
Eq (6): you do not define [ε]
Eq (8): [V] or [v] ?
Eq (9): using {v} for velocity vector and [v] for matrix of specific magnetic resistivity is quite confusing
Eq (12): define r and i subscripts
Eq (13): you sum from 1 to 1 ?
Eq (14): you use the same variable (n) as in Eq (13) with another definition. Please change the name.
93-96: this assumption + Eq (15) could be placed directly after Eq (9) ?
107: these forces are
110: is set to zero
112-113: is SOLID98 type obvious for the reader ?
Eq (16): ρ is now the density when it was charge density before. Please change variable name.
Eq (17): V is now velocity when it was the electric scalar potential before. Please change variable name.
118: is an electromagnetically induced
119-120: a turbulent mode (or a transitional one from laminar to turbulent) is present
120: In the case considered, the flow under investigation presents a Reynolds number around...
121: these equations were
121-122: The way to take turbulence into account in both cases is presented below.
122-123: through the manuscript, you use either "modeling" or "modelling". Please make a choice.
132: yields to
133: where overline means
138: where Cµ=... differs from standard k-ε model (it is not constant).
140-141: please explain why this cannot be used for pulsed MF ?
141 : you do not make any reference to Figure 3, the location of which is questionable since it refers to 2.1, 2.2 and 2.3. It would help understanding if you add a chapter 2.4 which would be a summary of your modeling approach, including this figure and a few comments on it.
151: 1...0.05 : I do not understand
152: In z direction element size was eqal to 1.3 mm (?)
153-155: in 2.2.1 you begin with boundary conditions. In 2.2.2 you finish with. Please homogenize.
154: ...by fluid flow are negligible.
163: please define Amush
164-166: you mix present and past. Please use present.
168: the thermophysical properties used are listed in Table 1
Table 1: density of liquid Ga : ro is missing and unit is not correct; for µ and σ, do not write the unit in the Property column.
169: Fluid dynamic and solidification...
171: see my remark on Figure 3 above
172 : all the chapter 2.1.1 should be put in chapter 2.4 Experimental Setup and Validation
179-180: the upper surface near the lateral wall region
180: you cannot refer to Figure 6 when the next Figure is 4... Please add a sketch (new Figure 4) to describe the apparatus.
195: along or against axis z
199: close to the axis of symmetry
Fig 4.b : please add unit B(T)
205-206: On the axis of symmetry of the vessel
Fig 6: to be put after Fig 5
215: you can note a rather good agreement between measurements and simulations, but it is not really a proof of convergence of the simulations
Fig 6b and 6c: it is rather unclear: does the blue/black colors presented in Fig 6c apply to Fig 6B too ? It seems right, but please precise then. What is the difference then between velocities in both Figure ? They are certainly not measured/computed at the same place, but from your explanations, I did not catch where curves of Fig 6c are taken from.
223: On the other hand, in the central region,
Fig 7: please precise what we see in the experimental part. It seems +/- related to liquid fraction ?
Fig 7: please precise in the legend that velocity vectors are superimposed to liquid fraction maps, since it is rather difficult to catch them at first sight
Fig 8: which is the location of the y axis that you choose ? Please precise
245: in some cases slightly differs
246: the fact that the LES differs from RANS is not surprising. Please precise why you choose LES, because "slightly differs" is not enough to prove that it is better.
249: at different pulsation frequencies fp are shown
255-256: close to the walls, excepted for the fp=0.1 Hz case, it is as you say rather difficult to conclude on experimental data. We see such a cloud of points that it is possible to see the numerical fluctuations that you superimpose, but also almost anything else... It could intersting to give some conclusions on the measurements (limits ?) and to calculate approximately the amplitude of fluctuations to prove that they exist at these frequencies.
Fig 10: would it be helpful to have the same scale on the four figures ?
256-257: would not it be more logical to state that the fluctuations of the liquid metal flow are equal to the magnetic field pulsation ?
Fig 11 fp=0.1 Hz: is there any interest in the fluctuation mode that you obtain in the simulations for instance at t=[21;23] or [31;33] s ? Can you explain ? It can also be observed experimentally, even if measurements are limited as you state.
288: define what Vch is
Fig 12: Dependencies
Fig 12: for the sake of hoogeneity, it would be better to represent experimental curves with dots and simulated ones with plain lines
289-290: precise Hz after numerical value of fm and fch
293-295: concerning the max value, especially the experimental curve, I would be more circumspect. It is difficult to identify a trend, and statistically I am not sure that you can reasonably prove that a line would not match
295: so-called
307: it is correlated
309: This review shows

Appendix A: the registered symbol is used only for Ansys Fluent. Is it normal ?
379-380: title of the paper should be lowercase

Author Response

Dear reviewer,

Thank you so much for all of your comments and help, I really appreciate that! I have made all the corrections in the manuscript, all changes are marked as pink.

  1. the strange organization of the chapters (The chapters and sections are reorganised as requested now. Thank you!)
  2. some approximations and lack of explanations (in the numerical model, in the description of figures, etc.) (all of the explanations in equations, tables and figures are added)
  3. a quite poor proof reading as far as I may note. Misprints are numerous, figures are not always located logically and their numerotation does not follow the right order, etc. (additional proof reading is done)

1: on a liquid metal

— done

11: it was found the ability : I do not catch the sentence

— sentence is changed

13: through the manuscript, you use either "behavior" or "behaviour". Please make a choice.

— thank you for noticing, corrected.

19: leads to the movement of what ?

— fluid movement

42 : the formation of segregations

— done

46: Hachani et al. show... : you switch from past to present (see also 58)

— corrected

Eq (9): using {v} for velocity vector and [v] for matrix of specific magnetic resistivity is quite confusing

— We have replaced [v] to [mu]-1 in this equation, because in my formulation there are no permanent magnets

112-113: is SOLID98 type obvious for the reader ?

— SOLID98 is the element from Ansys APDL Mechanical Element Library. We have added clarification in a footnote.

Eq (16): ρ is now the density when it was charge density before. Please change variable name.

— electric charge density changed to rho_e and the electrical resistivity changed to rho_r. Density remains the same.

Eq (17): V is now velocity when it was the electric scalar potential before. Please change variable name.

— electric scalar potential changed to V_e

118: is an electromagnetically induced

— done

119-120: a turbulent mode (or a transitional one from laminar to turbulent) is present

— In our case Reynolds number reaches ~10^5, which means we have a turbulent flow mode.

120: In the case considered, the flow under investigation presents a Reynolds number around...

— replaced

121: these equations were

— replaced

121-122: The way to take turbulence into account in both cases is presented below.

— replaced

122-123: through the manuscript, you use either "modeling" or "modelling". Please make a choice. 

— corrected

132: yields to

— done

133: where overline means

— done

138: where Cµ=... differs from standard k-ε model (it is not constant).

— changed

140-141: please explain why this cannot be used for pulsed MF ? 

— It can be used, but we prefer to use more precise LES approach. Sentence is corrected.

141 : you do not make any reference to Figure 3, the location of which is questionable since it refers to 2.1, 2.2 and 2.3. It would help understanding if you add a chapter 2.4 which would be a summary of your modeling approach, including this figure and a few comments on it.

— done

152: In z direction element size was eqal to 1.3 mm (?) 

— Yes, in the z direction, the distribution of the elements was uniform with an element size of 1.3 mm.

153-155: in 2.2.1 you begin with boundary conditions. In 2.2.2 you finish with. Please homogenize.

— The boundary conditions are the same for 2.2.1 and 2.2.2 as well. We moved description of boundary conditions to section 2.2

154: ...by fluid flow are negligible. 

— changed

163: please define Amush

— Mushy zone parameter is equal to 10^5.

164-166: you mix present and past. Please use present.

— corrected

168: the thermophysical properties used are listed in Table 1

— corrected

Table 1: density of liquid Ga : ro is missing and unit is not correct; for µ and σ, do not write the unit in the Property column.

— All corrections are taken into account; Table is modified by adding units column, to make it more readable.

169: Fluid dynamic and solidification...

— corrected

171: see my remark on Figure 3 above

— reference on Fig.3 exists and Figure located in created 2.4 section “Simulation coupling”

172 : all the chapter 2.1.1 should be put in chapter 2.4 Experimental Setup and Validation

— The sections is merged.

179-180: the upper surface near the lateral wall region

— changed

180: you cannot refer to Figure 6 when the next Figure is 4... Please add a sketch (new Figure 4) to describe the apparatus. 

— UDV transducers are sown in Figure 4 now.

195: along or against axis z

— corrected

199: close to the axis of symmetry

— changed

Fig 4.b : please add unit B(T) 

— added

205-206: On the axis of symmetry of the vessel

— thank you for the comment, however, both ways are correct.

Fig 6: to be put after Fig 5

— fixed

215: you can note a rather good agreement between measurements and simulations, but it is not really a proof of convergence of the simulations

— It is just agreement of 2 particular profiles in whole volume of circulated liquid, you are right. But this case of electromagnetic stirring of rectangular liquid volume is well studied as we presented in Appendix A1. Several authors report about weak azimuthal component of flow, this means that we have quasi two dimensional case. In a symmetrical two-vortex flow pattern the downward and upward flows are the most valuable to measure. Thus, we can make a conclusion that our fluid flow simulation results. All authors listed in Appendix A1 use from 1 to 4 UDV transducers to measure the liquid metal bulk flow. Based on this we choose 2 UDV sensors measuring setup. Nevertheless, we have changed “convergence” to “agreement”.

Fig 6b and 6c: it is rather unclear: does the blue/black colors presented in Fig 6c apply to Fig 6B too ? It seems right, but please precise then. What is the difference then between velocities in both Figure ? They are certainly not measured/computed at the same place, but from your explanations, I did not catch where curves of Fig 6c are taken from.

— (b) is a mean profiles which correspond to udv transducers placement on fig (a), whereas

(c) is transient velocity in the points at the melt height of 45 mm. The figure caption is changed and legends of graphs added as well.

223: On the other hand, in the central region, 

—corrected

Fig 7: please precise what we see in the experimental part. It seems +/- related to liquid fraction ? 

Yes, this is liquid fraction or liquid/solid phase. Blue (or green) corresponds to already solidified and from red to yellow is a liquid Ga. This is processed data from neutron radiography experiment and exactly corresponds to density. The explanation is added.

Fig 7: please precise in the legend that velocity vectors are superimposed to liquid fraction maps, since it is rather difficult to catch them at first sight

— description is added

Fig 8: which is the location of the y axis that you choose ? Please precise 

— The chosen location is the near wall area, the same as in Fig. 6b. The figure description is added.

245: in some cases slightly differs

— done

246: the fact that the LES differs from RANS is not surprising. Please precise why you choose LES, because "slightly differs" is not enough to prove that it is better. 

— Moreover, LES approach is more suitable for predicting low-frequency velocity fluctuations in two vortex flow patterns [Umbrashko, A.; Baake, E.; Nacke, B.; Jakovics, A. Modeling of the turbulent flow in induction furnaces. Metal Mater Trans B 2006, 37. doi:10.1007/s11663-006-0065-0.]

249: at different pulsation frequencies fp are shown 

— changed

255-256: close to the walls, excepted for the fp=0.1 Hz case, it is as you say rather difficult to conclude on experimental data. We see such a cloud of points that it is possible to see the numerical fluctuations that you superimpose, but also almost anything else... It could intersting to give some conclusions on the measurements (limits ?) and to calculate approximately the amplitude of fluctuations to prove that they exist at these frequencies. 

— First of all, the experimental and numerical data on graphs is reversed, and now the numerical curves are better seen. I am sorry, but I do not really understand which kind of limitations do you mean?

As it is written at this step just the spin-up behaviour is compared to show that pulsation frequency do not change it. The more precise analysis of this is provided in next section. Additionally FFT analysis of experimental graphs we have attached in this Author's Notes to Reviewer.

The comparison of approximately calculated amplitudes for all cases is given in Figure 12.

Fig 10: would it be helpful to have the same scale on the four figures ?

— added. the same scale is set on all figures.

256-257: would not it be more logical to state that the fluctuations of the liquid metal flow are equal to the magnetic field pulsation ? 

— This statement is done in section 3.2.3. Pulsation frequencies analysis.

Fig 11 fp=0.1 Hz: is there any interest in the fluctuation mode that you obtain in the simulations for instance at t=[21;23] or [31;33] s ? Can you explain ? It can also be observed experimentally, even if measurements are limited as you state.

— Yes, this is interesting phenomena. This, let me say, secondary fluctuations may be the result of a square shape of the vessel, azimuthal flow or interaction of two vortexes. And it will be better devote a separate study on this phenomena. the description is added.

288: define what Vch is

— definition is added

Fig 12: Dependencies

— done

Fig 12: for the sake of hoogeneity, it would be better to represent experimental curves with dots and simulated ones with plain lines

— done

289-290: precise Hz after numerical value of fm and fch

— done

293-295: concerning the max value, especially the experimental curve, I would be more circumspect. It is difficult to identify a trend, and statistically I am not sure that you can reasonably prove that a line would not match 

— Yes, you are right. unfortunately, we do not catch experimentally this extremum. The sentence is changed.

295: so-called

— done

307: it is correlated

— done

309: This review shows

— done

Appendix A: the registered symbol is used only for Ansys Fluent. Is it normal ? 

— The registered symbol is removed.

379-380: title of the paper should be lowercase

— done

Round 2

Reviewer 1 Report

Paper was corrected and can be accepted in the present form.

Reviewer 2 Report

Dear Authors,

Thank you for the extensive correction (sorry for that !). The paper sounds quite clearer now. A few misprints:

- 287: vortexes --> vortices 

- 406-407: no uppercase for the title of the paper.

Yours faithfully